# STRUCTDROP: A STRUCTURED RANDOM ALGORITHM TOWARDS EFFICIENT LARGE-SCALE GRAPH TRAINING

## ABSTRACT

Graph neural networks (GNNs) have gained considerable success in graph-based learning tasks, yet training GNNs on large graphs is still inefficient. The root cause is the graph-based sparse operations are difficult to accelerate with commodity hardware. Prior art reduces the computation cost of sparse matrix based operations (e.g., linear) via sampling-based approximation. However, two under-explored pain points still persist in this paradigm: ① *Inefficiency Issue.* The random-based sampling approaches have the non-zero entries randomly distributing over adjacency matrix, which slows down memory access process and is difficult to accelerate with commodity hardware. ② *Under-fitting Problem.* The previous sampling methods only utilize the same subset of nodes during the training, which may cause the under-fitting problem on other remain nodes. Aiming to systematically address these two pain points, we propose Structured Dropout, a.k.a, `StructDrop`. This method involves the selective random sampling of columns and rows from a sparse matrix for computation. Comprehensive experiments validate the efficiency and generalization of our framework: `StructDrop` achieves up to 5.09x speedup for a single sparse operation and 6.48x end-to-end speedup with negligible accuracy loss or even better accuracy.

## 1 INTRODUCTION

Graph Neural Networks (GNNs) have made significant advancements in various graph-related tasks (Hamilton et al., 2017; Hu et al., 2020; Ying et al., 2018; Jiang et al., 2022; Zhou et al., 2022; 2023). Yet, training GNNs can be time-inefficient, especially on large graphs. This is largely due to their two-phase execution: aggregation and update. During the aggregation phase, each node accumulates messages from its neighboring nodes using *sparse matrix-based operations*. Following this, in the update phase, nodes adjust their embeddings based on these messages through *dense matrix-based operations* (Fey & Lenssen, 2019; Wang et al., 2019). As shown in Figure 1, SpMM represents sparse operations and MatMul represents dense operations in their respective phases. It's notable that the aggregation phase can consume over 90% (Han et al., 2023) of the total GNN training time. This inefficiency stems from the nature of sparse matrix operations, which require numerous random memory accesses with minimal data reuse. Current hardware like CPUs and GPUs struggle to accelerate these processes (Duan et al., 2022b; Han et al., 2016; Duan et al., 2022a; Liu et al., 2023c). As a result, training GNNs on large-scale graphs is often very time consuming.

Research towards addressing this issue can be roughly grouped into three main categories. First, some works propose distributed GNNs training systems. These approaches aim to design GNN training platforms that reduce communication costs between hardware (Zheng et al., 2020; Ramezani et al., 2022; Wan et al., 2022b; Md et al., 2021; Wan et al., 2022a). Second, another line of research focuses on improving the efficiency of sparse operations; they achieve this by reducing memory access and combining consecutive operations (Zhang et al., 2022; Huang et al., 2020; Rahman et al., 2021; Wang et al., 2021). Third, some other works try to accelerate the training process from the optimization aspect, i.e., using fewer iterations to converge (Narayanan et al., 2022; Cong et al., 2020; Xu et al., 2021; Han et al., 2023; Jin et al., 2021; Cai et al., 2021).

In parallel, to cope with this challenge, previous work also tried to accelerate the sparse graph-based operation with randomized matrix multiplication (Liu et al., 2023a). To illustrate, consider a linear operation involving two matrices, $\mathbf{A} \in \mathbb{R}^{n \times m}$ and $\mathbf{B} \in \mathbb{R}^{m \times q}$. Initially, we obtain $\mathbf{A}' \in \mathbb{R}^{n \times k}$ and $\mathbf{B}' \in \mathbb{R}^{k \times q}$ ($k < m$) by selecting $k$ representative columns from $\mathbf{A}$ and their corresponding rows from $\mathbf{B}$, which we term as **column-row pairs** (Drineas et al., 2006). Following this, the matrix production $\mathbf{A}\mathbf{B} \approx \mathbf{A}'\mathbf{B}'$. With this procedure, the number of floating-point operations (FLOPs) and memory access are both reduced. This method sacrifices a degree of certainty to gain a reduction in computational complexity. Yet, only the sparse operations in the backward pass are replaced with their randomized counterparts to ensure the unbiasedness of gradient (Liu et al., 2023a;b). Given that every training step contains two such sparse operations - one forward and one backward - the maximum speedup from the prior method is limited to $2\times$ (Liu et al., 2023a).

In this paper, we further explore replacing the matrix multiplication in the forward pass with its randomized counterpart to obtain larger speedup. Prior research suggests the probability of choosing each column-row pair should be in proportion to the production of the respective row norm and column norm (Drineas et al., 2006). Interestingly, we observed that the column-row pairs selected in the forward pass exhibited a remarkable consistency across nearby iterations. We hypothesize that this consistency will cause under-fitting problem as they only utilize the same subset of nodes during training. Drawing from this insight, we propose a straightforward strategy: **the uniform selection of column-row pairs.** Namely, we assign the same probability to be sampled for each column-row pairs. Surprisingly, we found that this simple strategy can often outperform the complicated norm-based one in the graph learning problem. To further reduce the negative impact of the variance from uniform sampling, we propose to utilize instance normalization to stabilize the training process. In summary, our contributions are summarized as follows:

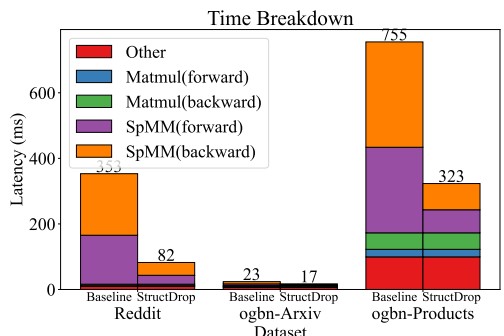

Figure 1: The time profiling of a three-layer GCNs on different datasets. SpMM may take $70{\sim}90\%$ of the total time. **Our method (StructDrop ) can reduce the total training time by** $6.48\times$. We measure the time on a NVIDIA A40 GPU. The detailed software and hardware information can be found in Appendix A.

- We observe that the norm-based column-row pairs selection tend to be the same across nearby iterations. We hypothesize that this consistency will cause under-fitting problem.

- Based on the observation, we suggest one simple strategy - uniform sampling. We show that the uniform sampling can beat the previous sampling methods in graph learning problems.

- Our approach can achieve 6.48x speedup with negligible accuracy loss or better accuracy.

## 2 PRELIMINARIES AND BACKGROUND

### 2.1 GRAPH NEURAL NETWORKS

We consider an undirected graph $G = (\mathcal{V}, \mathcal{E})$, where $\mathcal{V}$ and $\mathcal{E}$ denote the sets of nodes and edges, respectively, of size $N = |\mathcal{V}|$ and $E = |\mathcal{E}|$. Let $\mathbf{A} \in \mathbb{R}^{n \times n}$ denote the adjacency matrix, $\mathbf{A}_{i,j} = 1$ if $(v_i, v_j) \in \mathcal{E}$ else $\mathbf{A}_{i,j} = 0$, and let $\mathbf{X} \in \mathbb{R}^{n \times d}$ denotes the feature matrix. Based on the spatial message passing, GNNs learn the node representation through aggregating the neighbors' embeddings and combining with itself layer by layer. For example, the node embedding learning at the $l^{\text{th}}$ layer of Graph Convolutional Network (GCN) (Kipf & Welling, 2017) is defined as:

$$\boldsymbol{H}^{(l)} = \tilde{\boldsymbol{A}}\boldsymbol{X}^{(l-1)}\boldsymbol{W}^{(l)}, \boldsymbol{X}^{(l)} = \text{ReLU}(\boldsymbol{H}^{(l)}) \tag{1}$$

where $\boldsymbol{X}^{(l)} \in \mathbb{R}^{N \times d}$ is the node embedding matrix at the $l^{\text{th}}$ layer and $\boldsymbol{X}^{(0)} = \boldsymbol{X}$; $\tilde{\boldsymbol{A}} = \tilde{\boldsymbol{D}}^{-\frac{1}{2}}(\boldsymbol{A} + \boldsymbol{I})\tilde{\boldsymbol{D}}^{-\frac{1}{2}}$ is normalized adjacency matrix, $\tilde{\boldsymbol{D}}$ is the diagonal degree matrix of $\boldsymbol{A} + \boldsymbol{I}$; $\boldsymbol{W}^{(l)} \in \mathbb{R}^{d \times d}$ is trainable weight. In practice, $\tilde{\boldsymbol{A}}$ is often stored in sparse matrix format like compressed sparse row

(CSR) to save the computation cost (Fey & Lenssen, 2019). In each training step of backpropagation, it has exactly two phases, i.e., one forward phase and one backward phase. From the implementation perspective, its computation can be written as:

$$\text{Forward Pass} \qquad \boldsymbol{J}^{(l)} = \texttt{MatMul}(\boldsymbol{X}^{(l-1)}, \boldsymbol{W}^{(l)}), \boldsymbol{H}^{(l)} = \texttt{SpMM}(\tilde{\boldsymbol{A}}, \boldsymbol{J}^{(l)}), \qquad (2a)$$

$$\text{Backward Pass} \qquad \nabla\boldsymbol{J}^{(l)} = \texttt{SpMM}(\tilde{\boldsymbol{A}}^\top, \nabla\boldsymbol{H}^{(l)}) \qquad (2b)$$

$$\nabla\boldsymbol{X}^{(l-1)} = \texttt{MatMul}(\nabla\boldsymbol{J}^{(l)}, \boldsymbol{W}^{(l)}) \qquad (2c)$$

$$\nabla\boldsymbol{W}^{(l)} = \texttt{MatMul}(\boldsymbol{X}^{(l-1)\top}, \nabla\boldsymbol{J}^{(l)}), \qquad (2d)$$

where $\texttt{SpMM}(\cdot, \cdot)$ is the Sparse-Dense Matrix Multiplication and $\texttt{MatMul}(\cdot, \cdot)$ is the normal Dense-Dense Matrix Multiplication. From above, we can see that **each training step has exactly two SpMM operations.** In practice, although using a sparse matrix format can reduce computational cost compared to using a dense representation of the adjacency matrix, it is still notoriously inefficient on commodity hardware due to the cache miss problem (Han et al., 2016). As shown in Figure 1, we observed that $\texttt{SpMM}$ can take more than half of the training time.

## 2.2 FAST MATRIX MULTIPLICATION WITH SAMPLING

Given matrices $\boldsymbol{X} \in \mathbb{R}^{n \times m}$ and $\boldsymbol{Y} \in \mathbb{R}^{m \times q}$, our objective is to efficiently estimate the matrix product $\boldsymbol{XY}$. While the Truncated Singular Value Decomposition (SVD) provides an optimal low-rank estimation of $\boldsymbol{XY}$ (Adelman et al., 2021), its computational cost is almost equivalent to matrix multiplication. To overcome this issue, sampling algorithms have been proposed to approximate the matrix product $\boldsymbol{XY}$. The method involves sampling $k$ columns from $\boldsymbol{X}$ and their corresponding rows from $\boldsymbol{Y}$, resulting in smaller matrices. These matrices are then multiplied in the traditional manner (Drineas et al., 2006). Such an approach cuts down the computational complexity from $\mathcal{O}(mnq)$ to $\mathcal{O}(knq)$. Mathematically, the approximation is given by:

$$\boldsymbol{XY} \approx \sum_{t=1}^{k} \frac{1}{s_t} \boldsymbol{X}_{:,i_t} \boldsymbol{Y}_{i_t,:} = \texttt{approx}(\boldsymbol{XY}), \qquad (3)$$

where $\boldsymbol{X}_{:,i}$ and $\boldsymbol{Y}_{i,:}$ represent the $i^{\text{th}}$ column of $\boldsymbol{X}$ and the $i^{\text{th}}$ row of $\boldsymbol{Y}$, respectively. Within this context, we define the $(\boldsymbol{X}_{:,i}, \boldsymbol{Y}_{i,:})$ as the $i^{\text{th}}$ column-row pair. The term $k$ denotes the number of samples, and $s_t$ is a scale factor. $k$ is the number of samples ($1 \leq k \leq m$). $\{p_i\}_{i=1}^{m}$ is a probability distribution over the column-row pairs. $i_t \in \{1, \cdots m\}$ is the index of the sampled column-row pair at the $t^{\text{th}}$ trial. $s_t$ is the scale factor. (Drineas et al., 2006) indicates that setting $s_t = \frac{1}{kp_{i_t}}$ ensures the expectation the approximation is equal to the actual matrix multiplication results. Moreover, the approximation error is minimized when the sampling probabilities are proportional to the product of the norms of the column-row pairs:

$$p_i = \frac{||\boldsymbol{X}_{:,i}||_2 \, ||\boldsymbol{Y}_{i,:}||_2}{\sum_{j=1}^{m} ||\boldsymbol{X}_{:,j}||_2 \, ||\boldsymbol{Y}_{j,:}||_2}. \qquad (4)$$

Though the above sampling method effectively accelerates matrix multiplication (Drineas et al., 2006), its direct application to neural networks might not be optimal. This is because it overlooks the unique distribution of neural network weights. Observations indicate that neural network weight distributions tend to remain centered around zero during training (Glorot & Bengio, 2010; Han et al., 2015). Using this insight, (Adelman et al., 2021) introduced the **top-$k$ sampling** method: deterministically selecting the $k$ column-row pairs that have the highest values according to Equation 4, without any scaling. This equates to setting the probability $p_i$ of the top $k$ column-row pairs to 1, and to 0 for the others, with the scale factor $s_{i_t}$ being consistently 1.

Furthermore, (Liu et al., 2023a) adapted the top-k sampling technique to the domain of graph learning. To guarantee gradient unbiasedness, **they restricted the use of randomized matrix multiplication to the backward pass only, i.e., $\nabla\boldsymbol{J}^{(l)} = \textbf{SpMM}(\tilde{\boldsymbol{A}}^\top, \nabla\boldsymbol{H}^{(l)})$ in Equation 2b**. This decision was influenced by the understanding that the non-linear activation functions can alter the expected outcome of the approximated matrix multiplication (Liu et al., 2023a). While this approach preserves the final model accuracy, its potential for computational speedup is limited at $2\times$, given that it optimizes only the backward computations.

Table 1: Preliminary results on three datasets. "+Top-$k$ Sampling" means we replace both the forward and backward `SpMM` with their approximated version. Here we set the $k$ as $0.1|\mathcal{V}|$ across different layers. All reported results are averaged over six random trials.

|  |  | Reddit | ogbn-Arxiv | ogbn-Product |
|---|---|---|---|---|
| GCN | Baseline | $95.30 \pm 0.05$ | $72.09 \pm 0.26$ | $76.05 \pm 0.10$ |
| | +Top-$k$ Sampling | $93.53 \pm 0.44$ | $70.33 \pm 0.86$ | $74.73 \pm 1.81$ |
| GraphSAGE | Baseline | $96.59 \pm 0.03$ | $70.44 \pm 0.31$ | $78.05 \pm 0.90$ |
| | +Top-$k$ Sampling | $90.35 \pm 1.22$ | $62.10 \pm 0.52$ | $70.17 \pm 0.32$ |

In the next section, we explore the possibility of employing randomized matrix multiplication in the forward pass, aiming to achieve even higher acceleration.

## 3 METHODOLOGY

In this section, we first analyze why top-$k$ sampling cannot maintain the accuracy in Section 3.1. Specifically, we found that the top-$k$ sampling tends to select the same subset of nodes during the training, which may cause the under-fitting problem on other remain nodes. Based on this observation, we propose a simple yet effective strategy called `StructDrop` in Section 3.2, which uniformly select column-row pairs. Then in Section 3.3 we propose to incorporate instance normalization to further boost the stability of the training process with `StructDrop`.

### 3.1 MOTIVATION

Here we explore the possibility of accelerating the `SpMM` operations (Equation 2a) in both the forward and backward pass (Equation 2b). Specifically, we replace both the forward and backward `SpMM` with their approximated version using top-k sampling. Here we set the $k$ as $0.1|\mathcal{V}|$ across different layers. The model configuration is given in Appendix A. The performance is summarized in Table 1. We have observed a significant drop in performance, consistent with previous findings (Liu et al., 2023a). Upon closer examination, as shown in Figure 2, we found that the top-k sampling, as described in Equation 4, selects almost the same column-row pairs across nearby iterations. Namely, the Jaccard similarity across nearby iterations is around $90\%$. This indicates that the top-k sampling only utilizes the same subset of nodes during the training, which may cause the under-fitting problem on other remain nodes.

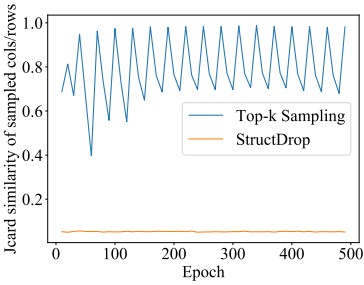

Figure 2: The Jaccard Similarity of selected column-row pairs across the iterations in Top-k Sampling.

To verify our hyperthesis, we plot the training accuracy and test accuracy of a three-layer GCN on ogbn-Products with different methods in Figure 3. The "under-fitting" hyperthesis is supported by Figure 3a, where the training accuracy with top-k sampling is noticeably lower than that of the baseline. As a consequence, Figure 3b shows that the test accuracy of GNNs trained with top-k sampling is also substantially inferior to the baseline.

In the next section, we explore how to cope with this under-fitting problem.

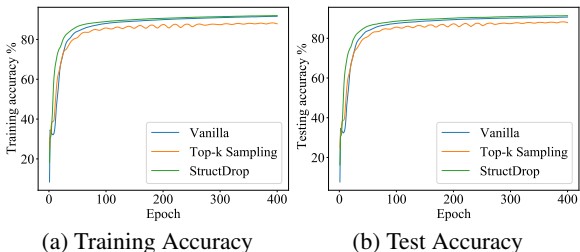

(a) Training Accuracy  (b) Test Accuracy

Figure 3: The training accuracy and test accuracy comparison between different methods, where we train a three-layer GCN on ogbn-Products

### 3.2 STRUCTDROP: DROP COLUMN-ROW PAIRS UNIFORMLY

Motivated by the observation that top-k sampling leads to under-fitting—stemming from the consistent selection of the same node subset throughout training—we introduce a straightforward strategy: uniformly selecting each column-row pair. Namely, **each column-row pair shares the same probability of being sampled, and we totally sample $k$ column-row pairs without replacement.** In this paper, we call this simple strategy `StructDrop` . Below we analyze the potential of our method from the generalization aspect.

**Generalization Analysis.** As shown in Figure 2, `StructDrop` utilizes different set of nodes during training. This suggests that `StructDrop` is effective in integrating information from the entire graph. `StructDrop` drops entire columns in the adjacency matrix while keeping the number of rows unchanged. It means that all of the out edges for a set of nodes are dropped out. The production over sampled adjacency matrix and node embeddings creates randomness during aggregation, and can be considered as a data augmentation mechanism. As a result, there is more randomness and deformations in the aggregated nodes, increasing generalizability. As a consequence, both Figure 3a and Figure 3b show that the training and test accuracy of `StructDrop` closely align those of the baseline. This indicates that `StructDrop` effectively addresses the under-fitting problem.

### 3.3 INSTANCE NORMALIZATION AT CRITICAL POSITION

Despite the prominent efficiency brought by the fast matrix multiplication with random sampling, a side effect is the significant distribution shift of node embeddings during training. In particular, given the random sampling of column-row pairs, the node embeddings are learned from the diverse sets of neighbors between epochs. It is widely found the acute distribution shift impedes learning rate and even misleads model to the convergence of poor-performing points Bjorck et al. (2018); Ioffe & Szegedy (2015); Bjorck et al. (2018).

To address the above issue, we propose to apply instance normalization right after the fast matrix multiplication. Mathematically, recalling the forward pass in Equation 2a, we use $\boldsymbol{H}^{(l)} = \text{SpMM}(\texttt{StructDrop}(\tilde{\boldsymbol{A}}, \boldsymbol{J}^{(l)}))$ denote the node embeddings after neighbor aggregation, which is obtained by uniformly dropping the column-row pairs over matrices $\tilde{\boldsymbol{A}}$ and $\boldsymbol{J}^{(l)}$ and conducting sparse matrix multiplication on them. Considering embedding $\boldsymbol{h}_i^{(l)} \in \mathbb{R}^d$ of node $v_i$ i.e., the $i^{\text{th}}$ row in $\boldsymbol{H}^{(l)}$, the instance normalization rescales it by Ulyanov et al. (2016):

$$\tilde{\boldsymbol{h}}_i^{(l)} = [\boldsymbol{h}_i^{(l)} - \text{E}(\boldsymbol{h}_i^{(l)})] \,/\, \text{Sqrt}(\text{Var}(\boldsymbol{h}_i^{(l)}) + \epsilon) * \boldsymbol{\gamma} + \boldsymbol{\beta}. \tag{5}$$

$\text{E}(\cdot)$, $\text{Sqrt}(\cdot)$, and $\text{Var}(\cdot)$ denote operations of expectation, squared root, and variance, respectively; $\boldsymbol{\gamma}, \boldsymbol{\beta} \in \mathbb{R}^d$ denote the trainable weights of running variance and mean, respectively. Each node embedding is rescaled to alleviate the sampling randomness and thereby converge model with superior generalization. The detailed experiments in discussing the node embedding shifting and generalization performance are provided in experiment section to support our proposals.

## 4 EXPERIMENTS

In our experiments, we evaluate our proposed framework through answering the following research questions: **Q1:** How well does `StructDrop` maintain accuracy with the reduced training time? **Q2:** How does the sampling ratio impact the final performance? **Q3:** What's the importance of instance normalization in the sampling scheme?

### 4.1 IMPLEMENTATION DETAILS

**Datasets, Backbones and Baselines** To evaluate `StructDrop` , we adopt four large scale graph benchmarks which are commonly used in different domains: Reddit (Hamilton et al., 2017), Reddit2 (Zeng et al., 2020) [1], ogbn-Arxiv (Hu et al., 2020) and ogbn-Products (Hu et al., 2020). We evaluate `StructDrop` using the full-batch training settings. We intergate `StructDrop` with two popular

---

[1]This is a sparser version of the original Reddit dataset ( 23M edges instead of  114M edges), and is used in paper GraphSAINT (Zeng et al., 2020)

Table 2: Comparison on the test accuracy on four datasets. The hardware for experiments is NVIDIA A40 (48GB). All results are averaged over six random trials.

| | | # nodes 232,965 | | # nodes 232,965 | | # nodes 169,343 | | # nodes 2,449,029 | |
| | | # edges 114,615,892 | | # edges 23,213,838 | | # edges 1,166,243 | | # edges 61,859,140 | |
| Model | Methods | Reddit | | Reddit2 | | ogbn-Arxiv | | ogbn-Products | |
| | | Accuracy | Speedup | Accuracy | Speedup | Accuracy | Speedup | Accuracy | Speedup |
| GCN | Vanilla | $95.3 \pm 0.05$ | $1 \times$ | $95.38 \pm 0.06$ | $1 \times$ | $72.09 \pm 0.26$ | $1 \times$ | $76.05 \pm 0.10$ | $1 \times$ |
| | Top-k Sampling | $93.21 \pm 0.15$ | $6.99 \times$ | $94.21 \pm 0.25$ | $2.72 \times$ | $70.84 \pm 0.63$ | $1.33 \times$ | $77.94 \pm 2.47$ | $1.96 \times$ |
| | DropEdge | $95.44 \pm 0.01$ | $1.87 \times$ | $95.47 \pm 0.02$ | $1.72 \times$ | $72.55 \pm 0.33$ | $1.21 \times$ | $78.96 \pm 0.60$ | $1.2 \times$ |
| | StructDrop | $95.47 \pm 0.05$ | $3.87 \times$ | $95.46 \pm 0.03$ | $2.4 \times$ | $72.46 \pm 0.23$ | $1.29 \times$ | $79.24 \pm 0.74$ | $1.8 \times$ |
| GraphSAGE | Vanilla | $96.59 \pm 0.03$ | $1 \times$ | $96.67 \pm 0.03$ | $1 \times$ | $70.44 \pm 0.31$ | $1 \times$ | $78.05 \pm 0.90$ | $1 \times$ |
| | Top-k Sampling | $92.73 \pm 0.33$ | $9.66 \times$ | $93.84 \pm 0.28$ | $3.08 \times$ | $63.75 \pm 0.42$ | $1.39 \times$ | $73.22 \pm 0.23$ | $3.31 \times$ |
| | DropEdge | $96.65 \pm 0.03$ | $2.65 \times$ | $96.55 \pm 0.03$ | $1.54 \times$ | $70.23 \pm 0.19$ | $0.81 \times$ | $78.57 \pm 0.09$ | $1.33 \times$ |
| | StructDrop | $96.65 \pm 0.04$ | $4.26 \times$ | $96.56 \pm 0.03$ | $2.33 \times$ | $70.03 \pm 0.26$ | $1.15 \times$ | $78.97 \pm 0.17$ | $2.47 \times$ |

models: GCN and GraphSAGE. Both models are trained with the whole graph at each training step. We use `SUM` aggregator in GCN and `MEAN` aggregator for GraphSAGE throughout this paper for fair comparison.

**Hyperparameter settings** `StructDrop` only has one hyperparameter, i.e., the sampling ratio. We present comprehensive sample ratio ablation study in Sec 4. We follow previous work Liu et al. (2023a) to sample the adjacency matrix every ten training steps, and reuse the sampled sparse matrix for multiplication within these ten steps. Due to the space limit, more experiments results regarding to this hyperparameter are included in the Appendix.

## 4.2 SUPERIOR EFFICIENCY AND GENERALIZATION OF `STRUCTDROP`

We start by evaluating the efficiency and accuracy of `StructDrop` comparing with different baselines sampling mechanisms. `StructDrop` accelerates the sparse operations in both forward and backward paths, thus largely reducing computational overhead. In the meantime, the accuracy of `StructDrop` is preserved, evidenced by the negligible accuracy loss or even better performance.

### 4.2.1 OPERATIONAL LEVEL ACCELERATION

Figure 1 shows the operation level performance gain of `StructDrop` . We measure the wall clock completion time of different operators on different datasets. With `StructDrop` , computation complexity in sparse matrix multiplication is largely reduced, so as leading to faster completion. Across datasets, the `SpMM` in forward path is accelerated by $1.9 \sim 5.5\times$, and `SpMM` in backward is accelerated by $2.62 \sim 4.8\times$. Overall, `StructDrop` achieves up to $5.09\times$ wall clock time speedup compare to the vanilla baseline.

### 4.2.2 END-TO-END PERFORMANCE ANALYSIS

Next we evaluate `StructDrop` 's end-to-end training time speedup as well as model accuracy comparing against different baselines. Specifically, we benchmark our approach against the standard training process without any approximations or accelerations. Additionally, we evaluate `StructDrop` alongside other sampling techniques such as Top-k sampling and DropEdge (Rong et al., 2019). We present results with the same sample ratio across to those sampled based baselines for fair comparison.

***Comparison against vanilla training scheme*** `StructDrop` achieves negligible accuracy loss (within 0.5%) or even better accuracy comparing to vanilla training scheme. The retained or boosted accuracy comes from `StructDrop` 's random sampling strategy during message aggregation phase. Such randomness acts as data augmentation, increasing the generalizability of `StructDrop` . We visualize the gap between training loss and testing loss on ogbn-Products dataset in Figure 3 to measure the generalizability of different mechanisms. The randomness and diversity in message aggregation leads to higher generalizability. In the meantime, `StructDrop` 's end-to-end wall clock training completion time achieves up to $6.48\times$ speedup compare to vanilla training scheme. To summarize, `StructDrop` is a new and effective acceleration scheme that makes GNN training process more efficient with retained accuracy. We next compare our training scheme with other sampling based mechanism.

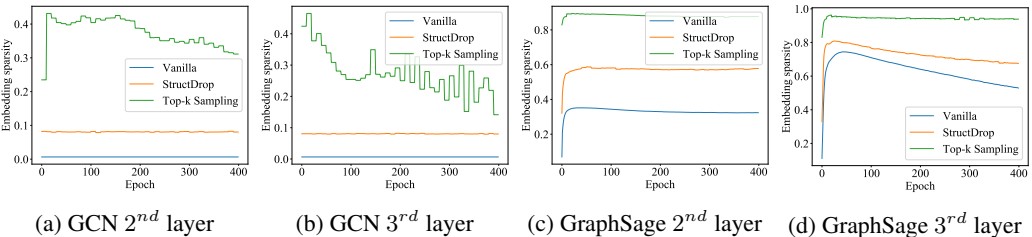

(a) GCN $2^{nd}$ layer     (b) GCN $3^{rd}$ layer     (c) GraphSage $2^{nd}$ layer     (d) GraphSage $3^{rd}$ layer

Figure 4: Embedding sparsity on different layers during training on Reddit2 dataset.

***Comparison against Top-$k$ sampling*** We now compare `StructDrop` with Top-k sampling. We highlight the accuracy gain of `StructDrop` over Top-$k$ sampling here. Top-$k$ sampling induces unacceptable performance loss compare to both vanilla baseline and `StructDrop` . This is because such Euclidean norm based sampling suffers from over-focusing on several columns and rows as analyzed in previous section, hence losing global graph information and leads to the underfitting behavior. On the other hand, random sampling in `StructDrop` is able to collect and utilize the global graph knowledge during message aggregation, thus achieves more comprehensive learning. Another important reason of poor performance in Top-k sampling is the information loss. We profile the embedding sparsity after message aggregation with vanilla, Top-k sampling and `StructDrop` in Figure 4. We found that after sampling and message passing, the embedding with Top-k sampling based mechanism has the highest zeros entry rate. Although Euclidean norm maintains minimal reconstruction error with vanilla sparse matrix multiplication, it tends to select columns/rows with lower degree (Liu et al., 2023a), which translates into higher sparsity and leads to larger embedding information loss in the aggregating phase, thus causing underfit. `StructDrop` doesn't have the concern of graph information loss. As we can see in Figure 4, the embedding sparsity of `StructDrop` and vanilla scheme is comparable on the other hand, leading to fewer information loss during message passing.

***Comparison against DropEdge*** DropEdge (Rong et al., 2019) is an effective method to alleviate overfitting and oversmoothing in graph neural network training. Similar to `StructDrop` , DropEdge randomly samples edges in the input graph basing on certain probability at each epoch. For fair comparison in this experiment, we set the dropping ratio to be 80% so that 20% of edges are retained, which is same with the `StructDrop` setting. Across different datasets, `StructDrop` achieves comparable accuracy(within 0.5%) with DropEdge, which reflects the effectiveness of data augmentation with sampled message passing. However, `StructDrop` achieves much higher efficiency comparing to DropEdge(up to 2.07x speedup), which mainly comes from the hardware efficiency. Though the number of edges preserved during training in each epoch is the same, the dropping granularity of Dropedge is smaller than `StructDrop` , that it only masks the value in adjacency matrix to zero. In sparse matrix multiplication, the matrix involved in calculation retains the same dimension. However, `StructDrop` randomly drops the entire columns and rows, leading to reduced adjacency matrix and node embeddings, which finally translates into lower level hardware performance gain while executing sparse matrix multiplication.

### 4.2.3 OVERFITTING AND GENERALIZATION

We have covered the overall performance comparison among `StructDrop` and different baselines in the previous section. Next, we want to understand more about the generalizability of `StructDrop` . We use ogbn-Products as an example to experimentally plot the training loss and generalization gap for different baselines and neural network architectures in Figure 5. The generalization gap is measured with the gap between training loss and testing loss. Thus higher generalization gap means better generalizability. Despite Top-k sampling mechanism with highest training loss and underfiting in training on GCN architecture, `StructDrop` achieves largest generalization gap. These results are aligning with our previous analysis, that the randomness and diversity caused by `StructDrop` can be served as data augmentation to increase the model generalizability.

Table 3: Accuracy and speedup regarding to different sample ratios on GCN architecture.

| Model | Ratio | Reddit | | Reddit2 | | ogbn-Arxiv | | ogbn-Products | |
|---|---|---|---|---|---|---|---|---|---|
| | | Accuracy | Speedup | Accuracy | Speedup | Accuracy | Speedup | Accuracy | Speedup |
| GCN | 0.1 | $95.44 \pm 0.04$ | $5.63 \times$ | $95.39 \pm 0.05$ | $2.81 \times$ | $72.16 \pm 0.21$ | $1.35 \times$ | $79.51 \pm 1.07$ | $2.04 \times$ |
| | 0.2 | $95.47 \pm 0.05$ | $3.87 \times$ | $95.46 \pm 0.03$ | $2.40 \times$ | $72.46 \pm 0.23$ | $1.29 \times$ | $79.24 \pm 0.74$ | $1.8 \times$ |
| | 0.3 | $95.47 \pm 0.04$ | $2.89 \times$ | $95.48 \pm 0.03$ | $2.05 \times$ | $72.44 \pm 0.24$ | $1.22 \times$ | $78.95 \pm 0.46$ | $1.6 \times$ |
| | 0.4 | $95.43 \pm 0.04$ | $2.26 \times$ | $95.46 \pm 0.04$ | $1.78 \times$ | $72.66 \pm 0.23$ | $1.17 \times$ | $78.63 \pm 0.29$ | $1.43 \times$ |

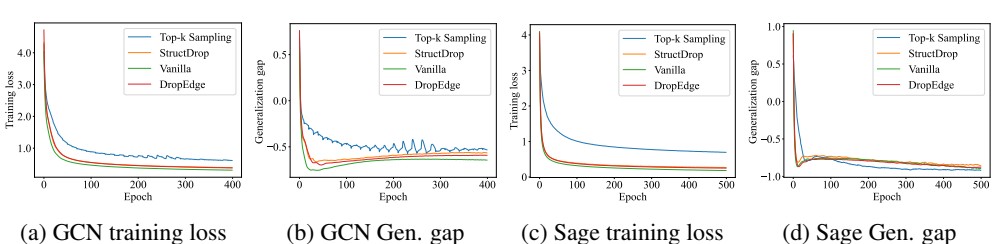

| (a) GCN training loss | (b) GCN Gen. gap | (c) Sage training loss | (d) Sage Gen. gap |
|---|---|---|---|

Figure 5: Training loss and generalization gap on ogbn-Products dataset

### 4.2.4 ABLATION STUDIES OF DROPPING RATIO

In this part, we presents comprehensive accuracy and speedup evaluation of `StructDrop`. Table 3 shows `StructDrop`'s results over different sampling ratios and datasets on GCN architecture. The effect of sample ratio with regards to accuracy are dependent to the datasets. For samll dataset like ogbn-Arxiv, it contains small number of edges. Higher sample ratio leads to higher accuracy as there will be less information loss. For obgn-Product with large number of edges, accuracy is reversed proportional to the sample ratio, as redundant edges causes the node embeddings to be smoothened by its neighbors, thus lose node features in its converged embedding. Regarding efficiency, lower sampling ratio leads to a higher computation speeds. The trend for GraphSAGE is similar to GCN, We delay the GraphSAGE results to the appendix section for more information.

### 4.3 BENEFITS FROM INSTANCE NORMALIZATION

We now evaluate the benefits brought by adapting instance normalization. Instance normalization acts as an distribution shifts mitigator that decrease the shifts of embeddings induced by random samplinng between epochs. Our experimental results show that instance normalization acts as effective factor that smooth the training process for better accuracy.

***Ablation study of instance normalization*** In this part, we evaluate the accuracy gain brought by instance normalization. We summarize the accuracy of GCN and GraphSAGE on different datasets w/o instance normalization. As shown in table 4, the accuracy with instance normalization applied is larger than that without instance normalization across different datasets. Instance normalization helps with `StructDrop` and leads to better accuracy on different architectures and datasets.

***Effect for smooth training*** Next we make a deep dive into the inner reason of instance normalization that helps boost the accuracy. We plot the distribution shift of the embedding after message aggregation with sampled columns/rows in 6. We use the norm of embedding difference between subsequent epochs to measure the smoothness of training. As shown in the Figure, training without instance normalization causes much larger embedding shifts between epochs, making the training

Table 4: Ablation study of instance normalization on different model architectures.

| | | Reddit | ogbn-Arxiv | ogbn-Products |
|---|---|---|---|---|
| GCN | w/ instance norm | $95.47 \pm 0.05$ | $72.46 \pm 0.23$ | $79.24 \pm 0.74$ |
| | w/o instance norm | $94.01 \pm 1.04$ | $69.30 \pm 1.19$ | $74.55 \pm 3.51$ |
| GraphSAGE | w/ instance norm | $96.65 \pm 0.04$ | $70.03 \pm 0.26$ | $78.97 \pm 0.17$ |
| | w/o instance norm | $96.52 \pm 0.04$ | $69.00 \pm 0.45$ | $78.25 \pm 0.21$ |

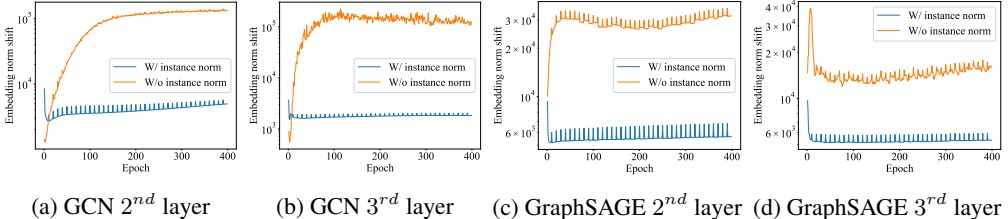

(a) GCN $2^{nd}$ layer    (b) GCN $3^{rd}$ layer    (c) GraphSAGE $2^{nd}$ layer    (d) GraphSAGE $3^{rd}$ layer

Figure 6: Embedding shift between different epochs on Reddit2 dataset

process not smooth as the model needs to constantly adapt to new inputs distribution, which is significant as the random samples causes message aggregation in different epochs varies drastically. Instance normalization successfully lowers the shifts between embeddings through different epochs, thus stablize the training process and leads to better accuracy.

## 5   RELATED WORK

**Large-scale Graph Learning.** Mathematically, the massage passing over graph could be described by sparse matrix multiplication, which is manipulated on graph adjacency and node embedding matrices. It is notorious that such message passing is resource consuming, where the memory and time complexities depend on the amounts of nodes and edges, respectively. To address the scalability issue on large graphs, numerous families of algorithms have been explored, including the subgraph-based GNN training Hamilton et al. (2017); Huang et al. (2018) , graph precomputation Wu et al. (2019); Klicpera et al. (2018); Yu et al. (2020), and distributed training Zha et al. (2023; 2022); Yuan et al. (2022); Wang et al. (2022). The common merit of them is to divide the large graph into many pieces, each of which could be handled by the resource-limited GPU.

**Efficient Training Algorithms.** Based on the above scalable training frameworks, another orthogonal line is to further reduce the memory and time consumption by approximating the message passing, which can be divided into following two categories. First, the adjacency matrix based approximation aims to compress the non-zero entries or matrix dimension. For example, Sketch-GNN proposes to sketch the graph adjacency matrix into a smaller one using hashing approach Chamberlain et al. (2022); DSpar expurgates the non-zero elements based on node degrees to obtain a sparse substitute Liu et al. (2023c). Second, the node embedding based approximation targets at compress the memory storage of hidden representations. For example, EXACT stocastically quantizes the node embeddings into low precision number Liu et al. (2022); GNNAutoScale stores the whole list of node embeddings in CPU and retrieve from it in forward propagation Fey et al. (2021).

**Random Dropout.** To improve the generalization performance on graph data, there are two main categories of dropout solutions. Edge-oriented dropout randomly samples a subset of edges at each epoch to avoid the over-fitting and over-smoothing, such as DropEdge Rong et al. (2019), Grand Feng et al. (2020b), etc. On the other hand, Node-oriented dropout removes node features and links connected to the dropped nodes. The node-oriented dropout approaches are originally motivated in sampling subgraph for scalable training and in augmenting graphs for contrastive learning, such as DropNode Feng et al. (2020a), FastGCN Chen et al. (2018), and LADIES Zou et al. (2019).

## 6   CONCLUSIONS

In this work we propose `StructDrop` to replace the time-consuming message passing with the fast and sparse matrix multiplication in forward and backward passes of GNNs. Particularly, `StructDrop` uniformly samples column-row pairs from graph adjacency matrix and node embedding matrix, reducing the computational complexity required in the sparse matrix multiplication `SpMM`. Furthermore, to address the distribution shift brought from random sampling, we propose to apply the instance normalization after `SpMM` to rescale node embeddings and stablize training dynamics. Extensive experiments on the benchmark datasets validate the effectiveness of our proposals in achieving the superior trade-off between efficiency and generalization performance.

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

# A  CONFIGURATION

Table 5: Configuration of Full-Batch GCN.

| Dataset | Training | | | Archtecture | | |
|---|---|---|---|---|---|---|
| | Learning Rates | Epochs | Dropout | BatchNorm | Layers | Hidden Dimension |
| Reddit | 0.01 | 400 | 0.5 | No | 3 | 256 |
| Reddit2 | 0.01 | 400 | 0.5 | No | 3 | 256 |
| *ogbn-Arxiv* | 0.01 | 500 | 0.1 | No | 3 | 512 |
| *ogbn-Products* | 0.001 | 400 | 0.5 | No | 3 | 256 |

Table 6: Configuration of Full-Batch GraphSAGE.

| Dataset | Training | | | Archtecture | | |
|---|---|---|---|---|---|---|
| | Learning Rates | Epochs | Dropout | BatchNorm | Layers | Hidden Dimension |
| Reddit | 0.01 | 400 | 0.5 | No | 3 | 256 |
| Reddit2 | 0.01 | 400 | 0.5 | No | 3 | 256 |
| *ogbn-Arxiv* | 0.01 | 500 | 0.1 | No | 3 | 512 |
| *ogbn-Products* | 0.001 | 500 | 0.5 | No | 3 | 256 |

# B  GRAPHSAGE ACCURACY AND EFFICIENCY

Table 7: Accuracy and speedup regarding to different sample ratios on GraphSAGE architecture

| Model | Ratio | Reddit | | Reddit2 | | ogbn-Arxiv | | ogbn-Products | |
|---|---|---|---|---|---|---|---|---|---|
| | | Acc. | Speedup | Acc. | Speedup | Acc. | Speedup | Acc. | Speedup |
| GraphSAGE | 0.1 | 96.53 ± 0.04 | 6.48 | 96.42 ± 0.04 | 2.93 | 68.83 ± 0.30 | 1.33 | 79.29 ± 0.07 | 2.96 |
| | 0.2 | 96.65 ± 0.04 | 4.26 | 96.56 ± 0.03 | 2.33 | 70.03 ± 0.26 | 1.15 | 78.97 ± 0.17 | 2.48 |
| | 0.3 | 96.69 ± 0.04 | 3.13 | 96.63 ± 0.04 | 2.01 | 70.35 ± 0.24 | 1.12 | 78.63 ± 0.12 | 2.1 |
| | 0.4 | 96.68 ± 0.02 | 2.42 | 96.67 ± 0.03 | 1.79 | 70.65 ± 0.34 | 1.06 | 78.31 ± 0.09 | 1.81 |

# C  CONVERGENCE ANALYSIS FOR TRAINING WITHOUT INSTANCE NORMALIZATION ON GCN

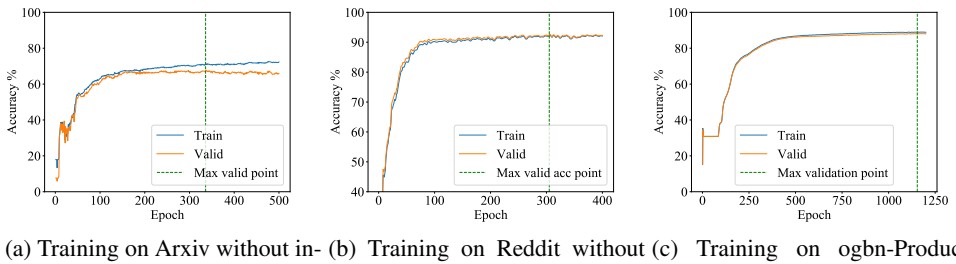

(a) Training on Arxiv without instance normalization

(b) Training on Reddit without instance normalization

(c) Training on ogbn-Product without instance normalization

Figure 7: Convergence analysis for different datasets

Table 8: The experiments for GCN2 architecture with different datasets and training mechanisms

| Dataset | Sample ratio | Vanilla | Speedup | Top-k | Speedup | StructDrop | Speedup | DropEdge | Speedup | DropNode | Speedup |
|---|---|---|---|---|---|---|---|---|---|---|---|
| Reddit2 | 0.1 | 96.80 ± 0.02 | 1 | 93.55 ± 0.07 | 2.32 | 96.65 ± 0.03 | 2.19 | 96.62 ± 0.02 | 1.96 | 96.21 ± 0.06 | 1.97 |
| | 0.2 | | 1 | 93.51 ± 0.58 | 2.1 | 96.72 ± 0.03 | 1.97 | 96.72 ± 0.01 | 1.6 | 96.31 ± 0.03 | 1.63 |
| ogbn-Arxiv | 0.1 | 72.12 ± 0.24 | 1 | 70.30 ± 0.32 | 1.26 | 71.52 ± 0.07 | 1.23 | 71.78 ± 0.23 | 1.16 | 71.76 ± 0.07 | 1.17 |
| | 0.2 | | 1 | 71.09 ± 0.09 | 1.21 | 72.16 ± 0.12 | 1.19 | 72.24 ± 0.30 | 1.13 | 72.35 ± 0.01 | 1.13 |
| Reddit | 0.1 | 96.81 ± 0.03 | 1 | 88.61 ± 0.83 | 6.29 | 96.72 ± 0.03 | 4.57 | 96.76 ± 0.03 | 3.16 | 96.24 ± 0.04 | 3.36 |
| | 0.2 | | 1 | 91.46 ± 1.00 | 5.14 | 96.82 ± 0.02 | 3.42 | 96.81 ± 0.07 | 2.01 | 96.39 ± 0.05 | 2.16 |

Table 9: The experiments for GraphSaint with different datasets and training mechanisms

| Dataset | Sample ratio | Vanilla | Speedup | Top-k | Speedup | StructDrop | Speedup | DropEdge | Speedup | DropNode | Speedup |
|---|---|---|---|---|---|---|---|---|---|---|---|
| ogbn-product | 0.1 | 78.67 ± 0.23 | 1 | 72.88 ± 0.13 | 1.4 | 79.42 ± 0.12 | 1.35 | 79.71 ± 0.14 | 0.58 | 79.47 ± 0.20 | 0.55 |
| | 0.2 | | 1 | 75.59 ± 0.37 | 1.32 | 79.59 ± 0.37 | 1.27 | 79.50 ± 0.18 | 0.53 | 79.27 ± 0.33 | 0.52 |
| reddit2 | 0.1 | 96.22 ± 0.05 | 1 | 87.31 ± 0.58 | 1.1 | 95.89 ± 0.01 | 1.1 | 95.90 ± 0.08 | 0.77 | 95.95 ± 0.07 | 0.76 |
| | 0.2 | | 1 | 91.27 ± 0.50 | 1.07 | 96.09 ± 0.03 | 1.05 | 96.12 ± 0.03 | 0.67 | 96.05 ± 0.11 | 0.68 |
| ogbn-arxiv | 0.1 | 70.72 ± 0.17 | 1 | 63.38 ± 0.20 | 1.13 | 68.94 ± 0.62 | 1.13 | 69.70 ± 0.23 | 0.84 | 68.70 ± 0.31 | 0.87 |
| | 0.2 | | 1 | 65.77 ± 0.41 | 1.11 | 69.40 ± 0.94 | 1.07 | 69.56 ± 0.06 | 0.79 | 69.47 ± 1.08 | 0.82 |
| reddit | 0.1 | 95.85 ± 0.13 | 1 | 87.30 ± 1.24 | 1.66 | 95.75 ± 0.08 | 1.47 | 95.81 ± 0.06 | 0.98 | 95.65 ± 0.03 | 0.98 |
| | 0.2 | | 1 | 90.36 ± 0.84 | 1.56 | 95.87 ± 0.05 | 1.35 | 95.92 ± 0.06 | 0.7 | 95.73 ± 0.08 | 0.73 |

Figure 7 specifically to check the convergence level with respect to the training epoch for training without instance normalization on GCN. As shown in the figure, the training of Reddit and ogbn-Arxiv without instance normalization has converged and the highest validation point is achieved far before the training ends. For ogbn-Product dataset, we plot a training curve with 1200 epochs (far more than commonly used configuration), and we see it converge very slowly during training without instance normalization. Note that the number we report in the Section 4.3 is following the configuration of experiments with instance normalization for ablation study. In ogbn-Product experiment, even with much larger epoch number for training, the accuracy finally achieves 76.73 ± 2.30, which is far less than with instance normalization. The results show that instance normalization does help with convergence speed as it contributes to the stabilized training under the randomized sampling training mechanism. Meanwhile, the smooth training process contributes to the final accuracy. Nonetheless, The variance of accuracy could be due to the instability of the training process, that because of the randomness the model converges to different points. With instance normalization, the smoother internal shifts (as shown in 6) lead to more robust training.

## D  EXPERIMENT FOR GCN2 ARCHITECTURE

Here we present another GNN architecture that is widely used — GCN2 for validating our results. The experiments compares `StructDrop` with different baselines performance on GCN2. Due to resource limitation, we didn't present the ogbn-Product experiment here. The results show a consistent conclusion with all other architectures, that `StructDrop` accelerates the sparse operations, and largely reduces the computation complexity while preserving the accuracy. The experiment results for GCN2 is shown in 8

## E  EXPERIMENT FOR GRAPHSAINT

Here we present GraphSaint experiment to validate whether we can accelerate the subgraph training mechanism effectively with `StructDrop`. Similar to previous experiments we compare different baselines. Through different baselines, `StructDrop` is the most Robust mechanism that it accelerate the training while maintaining the accuracy, which is consistent with other experiments. The experiment results for GraphSaint is shown in 9

Table 10: DropNode experiment for GCN and GraphSage

| Experiments | Dataset | Sample ratio | Vanilla | Speedup | StructDrop | Speedup | DropNode | Speedup |
|---|---|---|---|---|---|---|---|---|
| GCN | Reddit2 | 0.1 | 95.38 ± 0.06 | 1 | 95.39 ± 0.05 | 2.82 | 95.24 ± 0.02 | 2.32 |
| | | 0.2 | | 1 | 95.46 ± 0.03 | 2.40 | 95.35 ± 0.05 | 1.70 |
| | | 0.3 | | 1 | 95.48 ± 0.03 | 2.05 | 95.42 ± 0.03 | 1.40 |
| | | 0.4 | | 1 | 95.46 ± 0.04 | 1.78 | 95.41 ± 0.09 | 1.11 |
| | ogbn-Product | 0.1 | 76.05 ± 0.10 | 1 | 79.51 ± 1.07 | 2.05 | 76.31 ± 1.56 | 1.70 |
| | | 0.2 | | 1 | 79.24 ± 0.74 | 1.80 | 78.29 ± 2.15 | 1.17 |
| | | 0.3 | | 1 | 78.95 ± 0.46 | 1.60 | | |
| | | 0.4 | | 1 | 78.63 ± 0.29 | 1.44 | | |
| | ogbn-Arxiv | 0.1 | 72.09 ± 0.26 | 1 | 72.16 ± 0.21 | 1.36 | 71.99 ± 0.25 | 1.29 |
| | | 0.2 | | 1 | 72.46 ± 0.23 | 1.29 | 72.36 ± 0.20 | 1.23 |
| | | 0.3 | | 1 | 72.44 ± 0.24 | 1.22 | 72.54 ± 0.27 | 1.18 |
| | | 0.4 | | 1 | 72.66 ± 0.23 | 1.17 | 72.50 ± 0.35 | 1.13 |
| | Reddit | 0.1 | 95.30 ± 0.05 | 1 | 95.44 ± 0.04 | 5.64 | 95.21 ± 0.04 | 3.48 |
| | | 0.2 | | 1 | 95.47 ± 0.05 | 3.87 | 95.34 ± 0.06 | 2.07 |
| | | 0.3 | | 1 | 95.47 ± 0.04 | 2.89 | 95.32 ± 0.03 | 1.43 |
| | | 0.4 | | 1 | 95.43 ± 0.04 | 2.26 | 95.39 ± 0.03 | 1.08 |
| GraphSage | Reddit2 | 0.1 | 96.67 ± 0.03 | 1 | 96.42 ± 0.04 | 2.94 | 96.05 ± 0.02 | 2.19 |
| | | 0.2 | | 1 | 96.56 ± 0.03 | 2.33 | 96.33 ± 0.01 | 1.78 |
| | | 0.3 | | 1 | 96.63 ± 0.04 | 2.01 | 96.46 ± 0.04 | 1.50 |
| | | 0.4 | | 1 | 96.67 ± 0.03 | 1.79 | 96.53 ± 0.02 | 1.30 |
| | ogbn-Product | 0.1 | 78.05 ± 0.90 | 1 | 79.29 ± 0.07 | 2.96 | 78.30 ± 0.17 | 1.95 |
| | | 0.2 | | 1 | 78.97 ± 0.17 | 2.48 | 78.93 ± 0.20 | 1.32 |
| | | 0.3 | | 1 | 78.63 ± 0.12 | 2.10 | | |
| | | 0.4 | | 1 | 78.31 ± 0.09 | 1.81 | | |
| | ogbn-Arxiv | 0.1 | 70.44 ± 0.31 | 1 | 68.83 ± 0.30 | 1.33 | 68.81 ± 0.21 | 1.05 |
| | | 0.2 | | 1 | 70.03 ± 0.26 | 1.15 | 69.99 ± 0.29 | 1.02 |
| | | 0.3 | | 1 | 70.35 ± 0.24 | 1.12 | 69.87 ± 0.42 | 0.98 |
| | | 0.4 | | 1 | 70.65 ± 0.34 | 1.06 | 70.29 ± 0.14 | 0.94 |
| | Reddit | 0.1 | 96.59 ± 0.03 | 1 | 96.53 ± 0.04 | 6.48 | 96.11 ± 0.07 | 4.68 |
| | | 0.2 | | 1 | 96.65 ± 0.04 | 4.27 | 96.36 ± 0.06 | 2.72 |
| | | 0.3 | | 1 | 96.69 ± 0.04 | 3.13 | 96.50 ± 0.05 | 1.89 |
| | | 0.4 | | 1 | 96.68 ± 0.02 | 2.42 | 96.53 ± 0.06 | 1.43 |

# F  DROPNODE EXPERIMENT

We further conduct the experiments with DropNode across all different architectures with the same experimental setting. For fair comparison, we set the ratio of edge dropped to the same across different baselines in all experiments. The experiment results for DropNode is shown in 10. As shown in the results, DropNode experiment results share similar features with DropEdge, where StructDrop achieves a comparable accuracy with DropNode. Same as DropEdge, StructDrop achieves higher efficiency because of hardware efficiency as discussed in Section 4.2.2, StructDrop drops the entire columns and rows, leading to reduced adjacency matrix and node embeddings and finally translates into performance gain during training. The DropNode experiments for GCN2 and GraphSaint are shown in 8 and 9. Due to resource limitation, we didn't present the DropNode experiment for ogbn-Product datasets at sample ratio 0.3 and 0.4.

