# OpenReview forum: "STRUCTDROP: A STRUCTURED RANDOM ALGORITHM TOWARDS EFFICIENT LARGE-SCALE GRAPH TRAINING"
_ICLR.cc/2024/Conference — Submitted to ICLR 2024_

### Official Review · Reviewer_jrG5 · 2023-10-24

**Soundness:** 3 good
**Presentation:** 4 excellent
**Contribution:** 4 excellent
**Rating:** 6
**Confidence:** 3

**Summary:**

The authors propose a sampling mechanism for Graph Neural Networks (GNN) to improve its efficiency on commodity accelerators. The GNN training for large graphs is inefficient due to the requirement for two sparse matrix multiplications in the forward and backward passes of the gradient descent. The authors made an observation that sampling the row-column pairs of the adjacency matrix using their norms (as suggested by previous works since they provide the most accurate numerical approximation) leads to an under-fitting problem. The authors instead suggest sampling the row-column pairs uniformly and using instance normalization to stabilize the training. The experimental results and comparison with previous work show promising results for efficiency gains while retaining similar accuracy as the original GNN.

**Strengths:**

- The idea of structured sampling is simple and yet results in large performance gains with limited to none accuracy loss.
- The paper is well written and organized and can be followed easily by non-experts.
- The related literature has been sufficiently reviewed and nicely categorized.
- The authors provided sufficient ablation study on the effect of the instance normalization and dropping ratio.
- The authors have motivated the problem very well in the Introduction Section with examples.

**Weaknesses:**

- It was not clear why the authors chose only DropEdeg from the previous works on random dropout for GNN to compare with. It would’ve been better if they could compare StructDrop with more methods, especially the more recent ones like Grand [Feng et al. 2020b] and DropNode [Feng et al. 2020a].

**Questions:**

- It would be better if the authors could mention similar sampling methods like DropEdge in the introduction and explain the differences between the proposed method and them.
- It was not mentioned in the text that what numbers are reposted in Table 1, for example, are they the test accuracy?

---

> ### Author Response · Authors · 2023-11-22
> **Initial response to jrG5**
>
> We thank the reviewer for acknowledging the clarity of our proposed method, the delivery of ideas as well as the comprehensive experiments. We address the question for the reviewer below.
>
> # Response to “Adding experiment DropNode as a baseline”
>
> We thank the reviewer for raising this comment for enriching the evaluation and comparison. We further conduct the experiment of DropNode across all different architectures with the same experimental setting. For fair comparison, we set the ratio of edge dropped to the same across different baselines in all experiments. We present the experiment of DropNode in the appendix of the paper. As shown in the results, DropNode experiment results share similar features with DropEdge, where StructDrop achieves a comparable accuracy with DropNode. Same as DropEdge, StructDrop achieves higher efficiency because of hardware efficiency as discussed in section 4.2.2, StructDrop drops the entire columns and rows, leading to reduced adjacency matrix and node embeddings and finally translates into performance gain during training.
>
> # Response to “accuracy type in Table 1”
>
> The number reported in Table 1 is test accuracy during the experiment. We will add the clarification in paper to avoid ambiguity.

---

### Official Review · Reviewer_qoGN · 2023-10-29

**Soundness:** 3 good
**Presentation:** 3 good
**Contribution:** 3 good
**Rating:** 6
**Confidence:** 3

**Summary:**

This paper proposes StructDrop to speed up the sparse and fast matrix multiplication during the training of GNNs. The authors point out two main limitations in the previous works: the inefficiency issue and the under-fitting problem. To this end, they propose uniform sampling and instance normalization to address these problems. Experiments show that the proposed StructDrop achieves considerable speedup for sparse operations and end-to-end GNN with less to no or even better accuracy.

**Strengths:**

1. The authors observe the under-fitting problem from the previous column-row pairs selection and employ a uniform sampling to solve this problem to make the trained GNN models more generalizable and speed up the computation by setting the sample ratio.
2. To address the accuracy degradation problem from the fast matrix multiplication with sampling, this paper proposes an instance normalization to recover the accuracy for different graph model architectures.
3. The experiments, especially the ablation studies are well done and the paper is easy to follow.

**Weaknesses:**

1. In Table 4, without applying the instance normalization, the accuracy of GCN has a big difference between the max and min values, does this mean the convergence is not yet complete?
2. According to Table 2, for ogbn-Arxiv dataset, the DropEdge seems slower than vanilla algorithm, which means different datasets would achieve different speedups since their distribution, do you have any ideas about improving the sampling algorithm for specific datasets from their features?

**Questions:**

Please see above.

---

> ### Author Response · Authors · 2023-11-22
> **Initial response to qoGN:**
>
> We thank the reviewer for recognizing our contributions and writing. We address the questions raised below.
>
> # Response to “the large variance potentially caused by in-convergence for experiment without instance normalization in GCN”
>
> We thank the reviewer for the careful review! We add more experiments specifically to check the convergence level with respect to the training epoch. We added them to the appendix of our paper. As shown in the figure, the training of Reddit and ogbn-Arxiv without instance normalization has converged as the highest validation point is achieved far before the training ends. For ogbn-Product dataset, we plot a training curve with 1200 epochs (far more than commonly used configuration) and we see it converge very slowly during training without instance normalization. Note that the number we report in the paper is following the configuration of experiments with instance normalization for ablation study. In ogbn-Product experiment, even with 1200 epochs, the accuracy finally achieves 76.73 ± 2.30, which is far less than with instance normalization.
> The results show that instance normalization does help with convergence speed as it contributes to the stabilized training under the randomized sampling training mechanism. Meanwhile, the smooth training process contributes to the final accuracy. Nonetheless, The variance of accuracy could be due to the instability of the training process, that because of the randomness the model converges to different points. With instance normalization, the smoother internal shifts (as shown in figure 5) lead to more robust training.
>
> # Response to “different algorithms for fully utilizing the features of datasets”
>
> We thank the reviewer for the careful thoughts on designing the algorithm! We agree different datasets do have different features which could be utilized, for example, the information redundancy, the connectivity, etc, which could be considered in future work. For our proposed method, we used a general and uniformed sampling scheme, and we only have one hyper-parameter in configuration which is the sampling ratio. This method works well across different datasets.

---

### Official Review · Reviewer_4Sfp · 2023-10-31

**Soundness:** 2 fair
**Presentation:** 3 good
**Contribution:** 2 fair
**Rating:** 5
**Confidence:** 4

**Summary:**

This paper proposes StructDrop, a straightforward strategy that uniformly samples column-row pairs to optimize sparse matrix multiplication for accelerating graph training. They integrate the proposed strategy with the existing classic graph neural network (i.e., GCN and GraphSAGE) in both forward and backward passes. Experimental results show that the proposed approach achieves significant speedup in graph training compared to the vanilla baseline.

**Strengths:**

1. The proposed approach is simple and easy to follow. Experimental results show its effectiveness in accelerating graph training.
2. The paper is easy to read and generally well-written.

**Weaknesses:**

1. The theoretical foundation is insufficient. The proposed method appears naive and relies more on observation and intuition.
2. Experiments are insufficient. StructDrop only integrates with GCN and GraphSAGE. More classic models, such as GAT [1], and state-of-the-art models on large-scale graph data, like GraphSAINT [2], GCNII [3], Cluster-GCN[4], etc., should be included for comparison.

[1] Veličković, P., Cucurull, G., Casanova, A., Romero, A., Lio, P., & Bengio, Y. Graph attention networks. arXiv preprint arXiv:1710.10903, 2017.
[2] Zeng, H., Zhou, H., Srivastava, A., Kannan, R., and Prasanna, V. Graphsaint: Graph sampling based inductive learning method. In International Conference on Learning Representations, 2020.
[3] Chen, M., Wei, Z., Huang, Z., Ding, B., and Li, Y. Simple and deep graph convolutional networks. International Conference on Machine Learning, 2020. [4] Chiang, W. L., Liu, X., Si, S., Li, Y., Bengio, S., & Hsieh, C. J. Cluster-gcn: An efficient algorithm for training deep and large graph convolutional networks. In Proceedings of the 25th ACM SIGKDD international conference on knowledge discovery & data mining, 2019

**Questions:**

1. The proposed approach primarily relies on observation and intuition. More theoretical evidence is needed to explain why the proposed method is unbiased, how it ensures training accuracy, and its error boundaries.

2. The Top-k sampling method only accelerates the backward process, and the maximum speedup is limited to 2x. However, from Table 2 we can see the acceleration effect of Top-k sampling far exceeds that of StructDrop. Please explain the reasons.

3. It is insufficient to demonstrate the effectiveness of the proposed approach by only applying StructDrop to GCN and GraphSAGE. It would be helpful to integrate StructDrop with more classical GNN models, such as GAT [1].

[1] Veličković, P., Cucurull, G., Casanova, A., Romero, A., Lio, P., & Bengio, Y. Graph attention networks. arXiv preprint arXiv:1710.10903, 2017.

4. Both GCN and GraphSAGE do not perform well on large-scale data. Many GNN-related approaches have demonstrated better results and faster training speed on large graphs, such as GraphSAINT [2], GCNII [3], Cluster-GCN[4], etc. It would be beneficial to apply the proposed approach on these models for comparison.

[2] Zeng, H., Zhou, H., Srivastava, A., Kannan, R., and Prasanna, V. Graphsaint: Graph sampling based inductive learning method. In International Conference on Learning Representations, 2020.
[3] Chen, M., Wei, Z., Huang, Z., Ding, B., and Li, Y. Simple and deep graph convolutional networks. International Conference on Machine Learning, 2020. [4] Chiang, W. L., Liu, X., Si, S., Li, Y., Bengio, S., & Hsieh, C. J. Cluster-gcn: An efficient algorithm for training deep and large graph convolutional networks. In Proceedings of the 25th ACM SIGKDD International Conference on knowledge discovery & data mining, 2019

5. In Table 7, why does the accuracy decrease as the sample ratios increase in the ogbn-Products dataset?

---

> ### Author Response · Authors · 2023-11-22
> **Initial response to 4Sfp**
>
> # Response to “theoretical guarantee to StructDrop”
>
> We thank the reviewer for considering the overall soundness of our proposed scheme. Here we would like to argue that our work takes an initial step towards exploring random algorithms applying to the forward path. We report and find that sound theory for top-k sampling, which is an unbiased estimation, might not be the most robust algorithm for sampling in forwarding, which is inconsistent with intuition. We then experimented and proposed this effective random sampling algorithm, which works well for the training pipeline. We have done comprehensive analysis and experiments for verifying our observations. We hope this first step could benefit the community and innovate more work alongside this direction.
>
>
> # Response to “sampling efficiency comparing to top-k sampling in Table 2”
>
> We thank the reviewer for the careful reviewing our evaluation results. We would like to point out that for all the baseline listed in Table 2, they are conducted and compared in a fair way, that top-K sampling is also applied both in forward and backward pass. Among different mechanisms, Our conclusion is that StructDrop is the robustest in acceleration while preserving the accuracy.
>
> # Response to “adding more architecture for graph training”
> We thank the reviewer for the careful consideration. We would like to point out that GCN and GraphSAGE are still the robustest and most widely used GNN architectures in graph training. We add another GNN architecture that is widely used — GCN2 for validating our results. We add experiments of comparing different baselines on this architecture and we add the results in the appendix of the paper. The results show a consistent conclusion with all other architectures, that StructDrop accelerates the sparse operations, largely reducing the computation complexity while preserving the accuracy. Please refer to the appendix for more details. We believe the architecture presented in our paper are used in mainstream GNN training and our results are representative and convincing in GNN accelerations.
>
>
> # Response to “adding more graph training mechanisms”
>
> We thank the reviewer again for the careful consideration. As suggested, we add Graphsaint experiment to validate whether we can accelerate the subgraph training mechanism effectively using StructDrop. We add experiments of comparing different baselines on this mechanism and we add the results in the appendix of the paper. Comparing through different baseline, StructDrop is the most Robust mechanism that it accelerate the training while maintaining the accuracy, which is consistent with other experiments. Please refer to the appendix for more details.
>
>
> # Response to “Why does training accuracy decreased with the increased  sample ratio for ogbn-Product dataset”
>
> We thank the reviewer for the detailed review of this experimental results. We found that ogbn-Product’s performance decreased with the increased sample ratio. We further conduct experiments to change the dropout ratio (a training configuration) in ogbn-Product dataset training. We found consistent results with the results presented in the paper, that with larger dropout ratio, the accuracy decreases more. We explain the phenomenon as follows: the augmentation and generalization performance varies across different datasets. For ogbn-Product, it has much more edges/nodes compared to other datasets, leading to much more information redundancy. The augmentation scale, which can be defined by the ratio of dropped information, i.e. the dropping number of columns row pairs in our settings, will contribute more to the generalization ability, and improve the model’s performance as a result.

---

### Official Review · Reviewer_hFvm · 2023-10-31

**Soundness:** 3 good
**Presentation:** 3 good
**Contribution:** 2 fair
**Rating:** 3
**Confidence:** 4

**Summary:**

This work introduces Structured Dropout, i.e., StructDrop, to improve the efficiency of graph neural networks’ (GNNs) training on large graphs. Specifically, StructDrop replaces the sparse matrix multiplication (SpMM) in both the forward and backward passes of GNNs with its randomized counterpart achieved through uniformly sampling column-row pairs. Furthermore, to address the distribution shift brought by random sampling, instance normalization is applied after SpMM to rescale node embeddings and stabilize training. Experimental results on the benchmark datasets show that StructDrop significantly accelerates training with negligible accuracy loss or even better accuracy.

**Strengths:**

The proposed method, StructDrop, is straightforward and easy to understand. Experiments on benchmark datasets, employing two different GNN architectures, validate the effectiveness of StructDrop in accelerating the training of GNNs. The paper also demonstrates the benefits of incorporating instance normalization to mitigate the negative impact caused by StructDrop. Furthermore, the clarity and smooth flow of the paper contribute to its overall quality.

**Weaknesses:**

The proposed method, StructDrop, makes an incremental technical contribution within the context of existing research. Previous work has already explored the application of randomized matrix multiplication to sparse operations in the backward pass of GNNs. StructDrop builds upon this work by extending the method to the forward pass, with the primary modification being the adoption of a uniform sampling strategy for selecting column-row pairs, as opposed to the previous top-k sampling method. Furthermore, there are some inconsistent statements in this paper. For example, this paper states that StructDrop can address the inefficiency issue in the abstract but lacks elaboration in subsequent sections.

**Questions:**

1.	In the abstract, the paper highlights the inefficiency issue associated with random-based sampling approaches but lacks elaboration in subsequent sections.

2.	In section 2.2, the paper reviews fast matrix multiplication with sampling. In the original formulation, the column-row pairs are sampled based on the probability distribution given in Equation 4. Did you try this original probability distribution instead of uniform sampling and top-k sampling? It's better to add it as a comparison baseline.

---

> ### Author Response · Authors · 2023-11-22
> **Initial response to reviewer hFvm:**
>
> # Response to “incremental contribution”
> We claim our contribution as the first to employ random sampling during the whole training pipeline which largely accelerate the GNN training. We study and implement different sampling algorithms and effectively accelerate the graph training. We report and find that sound theory for top-k sampling might not be the most robust algorithm for sampling in forwarding. Instead, a pure random sampling algorithm could work the best. We proposed to apply instance normalization in this scheme for stabilizing the training. We also conduct comprehensive analysis and experiments to verify our results that we accelerate the large graph training far more than previous SOTA GNN acceleration[1] while preserving the accuracy.
>
> # Response to “Lack of elaboration for efficiency of random sampling based mechanism”:
> We would like to point out that our method has largely increased the efficiency of GNN training.  Our evaluation in Table 2 shows that our proposed method achieved at most 3.87x than the vanilla scheme while maintaining the accuracy. Moreover, we accelerated much more in large graph training then previous work. We are the first to study and apply sampling algorithms during the whole training process, and exceed the previous SOTA solution by more than 2x [1].
>
> # Response to “adding the original formula as part of our baseline”:
>
> We would like to point our in section 2.2 we wrote “(Adelman et al., 2021) introduced the top-k sampling method: deterministically selecting the k column-row pairs that have the highest values according to Equation 4”. In another word, top-k sampling is based on the equation 4 and is a better mechanism for forming up the sampling. We have thoroughly studied the behavior of top-k sampling and compare our results with it in section 4 as a baseline. Our conclusion is that top-k sampling cannot maintain the accuracy due to underflow, and applying StructDrop will achieve efficiency while preserving the accuracy.
>
>
>
> [1] Liu, Zirui, Chen Shengyuan, Kaixiong Zhou, Daochen Zha, Xiao Huang, and Xia Hu. "RSC: accelerate graph neural networks training via randomized sparse computations." In International Conference on Machine Learning, pp. 21951-21968. PMLR, 2023.

---

### Meta-Review · Area_Chair_Ugnc · 2023-12-05

**Metareview:**

The  paper presents StructDrop, an  approach designed to enhance the efficiency of training GNNs. This method, based on uniform sampling of column-row pairs and instance normalization, claims to significantly accelerate training while maintaining accuracy. The experimental results are compelling, showcasing StructDrop's capability to speed up training processes across benchmark datasets. However, the paper's contribution is viewed as somewhat incremental, primarily building upon previous work in randomized matrix multiplication and extending it to the forward pass of GNNs.

**Justification For Why Not Higher Score:**

There is a lack of comprehensive theoretical support and in-depth comparative analysis with other advanced GNN architectures and sampling methods.

**Justification For Why Not Lower Score:**

N/A

---

### Decision · Program_Chairs · 2024-01-16

Reject